# Lung Cancer Screening, towards a Multidimensional Approach: Why and How?

**DOI:** 10.3390/cancers11020212

**Published:** 2019-02-12

**Authors:** Jonathan Benzaquen, Jacques Boutros, Charles Marquette, Hervé Delingette, Paul Hofman

**Affiliations:** 1Department of Pulmonary Medicine and Oncology, Université Côte d’Azur, CHU de Nice, FHU OncoAge, 06100 Nice, France; benzaquen.j@chu-nice.fr (J.B.); boutros.j@chu-nice.fr (J.B.); 2Institute of Research on Cancer and Ageing (IRCAN), Université Côte d’Azur, FHU OncoAge, CNRS, INSERM, 06107 Nice, France; hofman.p@chu-nice.fr; 3Asclepios Project Team, Sophia Antipolis-Mediterranee Research Centre, Université Côte d’Azur, FHU OncoAge, Inria, 06902 Sophia Antipolis, France; herve.delingette@inria.fr; 4Laboratory of Clinical and Experimental Pathology and Biobank BB-0033-00025, Université Côte d’Azur, FHU OncoAge, CHU de Nice, 06001 Nice, France

**Keywords:** lung cancer, artificial intelligence, screening

## Abstract

Early-stage treatment improves prognosis of lung cancer and two large randomized controlled trials have shown that early detection with low-dose computed tomography (LDCT) reduces mortality. Despite this, lung cancer screening (LCS) remains challenging. In the context of a global shortage of radiologists, the high rate of false-positive LDCT results in overloading of existing lung cancer clinics and multidisciplinary teams. Thus, to provide patients with earlier access to life-saving surgical interventions, there is an urgent need to improve LDCT-based LCS and especially to reduce the false-positive rate that plagues the current detection technology. In this context, LCS can be improved in three ways: (1) by refining selection criteria (risk factor assessment), (2) by using Computer Aided Diagnosis (CAD) to make it easier to interpret chest CTs, and (3) by using biological blood signatures for early cancer detection, to both spot the optimal target population and help classify lung nodules. These three main ways of improving LCS are discussed in this review.

## 1. Introduction

Lung cancer (LC) is the leading cause of death from cancer, but early-stage treatment improves LC prognosis. The National Lung Screening Trial (NLST) demonstrated that annual LC screening (LCS) with low-dose computed tomography (LDCT) reduced mortality by 20% compared to controls [1] (Table 1). More recently, the Dutch–Belgian NELSON lung cancer screening trial presented in September 2018 at the International Association for the Study of Lung Cancer (IASLC) 19th World Conference on Lung Cancer (WCLC) in Toronto, Canada, showed reduced mortality by more than 25% in the LDCT arm compared to the control arm [2] (Table 1). Based on the NLST results, the United States Preventive Services Task Force (UPSTF) issued recommendations for LCS of people meeting the NLST criteria. The Centers for Medicare & Medicaid Services (CMS) decided to provide coverage for LCS in smokers aged 55 to 77 years with more than a 30-pack-years smoking history and who had not quit within the last 15 years [3,4]. Low-dose computed tomography is now the cornerstone of LCS in North America and Australia. Given the confirmatory results of the NELSON screening trial [2], it can be assumed that LDCT screening will be approved in Europe and that health authorities will very soon provide coverage for LDCT-based LCS [5,6], as for breast cancer (mammography) and colon cancer (colonoscopy). However, despite Medicare and Medicaid coverage, the take-up of LCS in the US remains very low (i.e., below 4%) [7,8,9]. The reasons for such a low take-up of LCS include: (1) patients not wanting screening (fatalism mentality in the elderly, stigma associated with LC, poor lifestyle choice); (2) patients’ awareness (i.e., less than breast cancer screening); (3) physicians not referring (difficult recall of smoking history, controversies among primary care societies, controversies among health agencies); and (4) a high false-positive rate requiring cumbersome follow-up [8,9,10]. Among these reasons, some are related to the practicality of LCS. In this respect, the need for repeated imaging and downstream diagnostic evaluations related to a high false-positive rate of LDCT (ranging from 26 to 58%) [1,7] is responsible for needless anxiety of patients and their family. In the Veterans Health Affairs (VHA) study, up to 52% of the screened patients who did not have LC required downstream diagnostic procedures [7].

The global shortage of radiologists facing a growing and aging population in Europe will quickly overload existing LC clinics and multidisciplinary teams. In addition, the high rate of false-positive results will lead to cumbersome follow-up and surveillance of incidental pulmonary nodules. Thus, there is urgent need to improve LDCT-based LCS, and especially to reduce the false-positive rate that plagues the current detection technology, to provide patients earlier access to life-saving intervention.

## 2. Lung Cancer Screening Can Be Improved

Lung cancer screening can be improved in several ways: (1) refine selection criteria (risk factor assessment); (2) use Computer-Aided Diagnosis (CAD) to make it easier to interpret chest CTs; (3) use biomarkers to detect early-stage LC, to spot the optimal target population or to help classify lung nodules; and (4) use highly sensitive bronchoscopic techniques to enhance the detection rate of central airway lesions.

### 2.1. Refine Selection Criteria to Improve the Effectiveness and Efficiency of Lung Cancer Screening

The following terms must be defined here: 1) screening effectiveness, which is the number needed to screen (NNS) per LC death prevented, and 2) screening efficiency, which is the number of false-positive results and downstream diagnostic procedures per LC death prevented (a surrogate of harm-to-benefit ratio).

Risk-based selection improved LCS effectiveness by 17% as compared to UPSTF screening criteria [11]. A similar conclusion was drawn by Caverly et al., who relied on the Bach risk model [12,13] in the VHA study and found an NNS per LC death prevented ranging from 687 in the highest risk quintile to 6903 in the lowest risk quintile [12]. Risk-based selection can also improve the LCS efficiency. In the VHA study, although the harm (overall rates of false positives requiring tracking or requiring downstream evaluations) did not differ between low- and high-risk quintiles, LCS was much more efficient in the high-risk quintile [7,12].

### 2.2. Use Computer Aided Diagnosis for Low-Dose Computed Tomography Interpretation to Facilitate Lung Cancer Screening and Lessen the False-Positive Rate

The interpretation of LDCT may be difficult in the setting of LCS. The simple algorithm design should be based on two questions surrounding the key lesion detected with LDCT (i.e., lung nodule): (1) “Does this individual have a nodule?” If the answer is “no”, then he/she will be given an appointment for the next screening round; (2) if the answer is yes, then the second question is “Is this nodule cancerous?” Depending on its features, the nodule will be classified as malignant (M), benign (B), or indeterminate (I) (Figure 1). Figure 2 exemplifies the range of difficulties encountered by physicians of LC clinics in the setting of LCS. Lung cancer screening takes time when relying on LDCT alone. Indeed, most decision-making algorithms for lung nodules advocate a repeat CT to study the volume-doubling time (VDT), a datum which, combined with the morphology of the nodule, has the most determinant weight to decide whether or not to go to invasive procedures including surgery [14].

To exemplify the stress and anxiety generated by the discovery of a lung nodule, one can look at the VHA experience in which 56% of the nonmalignant nodules required tracking and took an average of more than a year for the patient to be reassured (or not) of the nature of their nodule.

Deep convolutional neural networks (CNNs) have been successfully developed in the field of medical image analysis over the past five years and can be specifically trained for lung nodule detection and reduction of false positivity. CAD systems for LCS involve two steps: (1) detection of pulmonary nodules (which often includes lung segmentation, nodule detection, and segmentation); and (2) diagnosis of their malignancy based on the analysis of a set of features, such as volume, shape, VDT, and density gradient of each nodule (Figure 3). Currently, there are many studies about the first step, but few about the second step [15,16]. Convolutional neural networks are trained on publicly available databases (Table 2) and then tested on different datasets.

Training databases include: (1) chest CTs with annotated nodules, such as the Lung Nodule Analysis 2016 (LUNA16) dataset, which is a collection of 888 axial CT scans of the patients’ chest cavities taken from the Lung Image Database Consortium image collection (LIDC/IDRI) database [17]; in total, 1186 nodules were annotated across 601 patients; and (2) chest CTs labeled as “with cancer” if the associated patient was diagnosed with cancer within one year of the scan, and “without cancer” otherwise. Once trained, the CNN output provided a probability of malignancy between 0 and 1 (Figure 4).

The 2017 Kaggle Data Science Bowl was a critical milestone in support of the National Cancer Institute Cancer Moonshot by convening the data science and medical communities to develop LCS algorithms [18]. Using a dataset of 2101 high-resolution lung scans provided by the National Cancer Institute and labeled as “with” or “without cancer”, the 1972 competing teams have developed algorithms to accurately determine when lesions in the lungs were cancerous. Liao et al. won this 2017 Data Science Bowl by proposing the first volumetric end-to-end 3D CNN for 3D lung nodule detection and characterization with an AUC of 87% on the blinded test set [19].

### 2.3. Use Blood Biomarkers in the Setting of Lung Cancer Screening

Different tumor-derived components can be detected and isolated from blood samples, including circulating tumor cells (CTCs), circulating cell-free tumor DNA (cftDNA), cell-free tumor RNA (cftRNA), exosomes, and tumor-educated platelets (TEP) [20,21] (Figure 5). These components can be used as biomarkers: (1) to detect early stage LC; (2) to spot the optimal target population for LCS; or (3) to help classify indeterminate lung nodules.

#### 2.3.1. Biomarkers to Detect Early-Stage Lung Cancer

We previously showed that in high-risk patients (i.e., Chronic Obstructive Pulmonary Disease (COPD) and heavy smokers), circulating tumor cells (CTCs) detected with the isolation by size of epithelial tumor cell (ISET) technique (RARECELLS, Paris, France) could be detected in patients with COPD without clinically detectable LC up to four years before LC was identified on LDCT [22]. The CTCs detected had a heterogeneous expression of epithelial and mesenchymal markers, and some specific antigens (such as TTF1), which were similar to the corresponding phenotype of the lung tumor [23]. No CTCs were detected in control smoking and nonsmoking healthy individuals. From these preliminary results, we demonstrated for the first time that in high-risk patients, CTCs can be detected very early in the course of LC. We therefore launched a national prospective cohort study (the AIR study) to assess the role of CTCs in LCS in a high-risk population, that is, patients with COPD, heavy smokers, and >55-year-old patients (NCT02500693) [24].

In addition to CTCs, a more or less complex signature of the plasma microRNA (miRNA) has been shown to be associated with localized or metastatic LC [25,26,27]. More recently, studies have been performed on populations with a high risk of developing LC but without a known cancer. In particular, Sozzi et al. identified a signature of plasma microRNAs that showed an excellent predictive value for LC in a high-risk population [28]. In this latter study, the authors showed that the addition of a 24-microRNA signature classifier (MSC) to LDCT could raise LC detection sensitivity to 98% [28]. 

Montani et al. identified an LC-predictive signature of 13 microRNAs for high-risk individuals with a sensitivity of 77.8% and a negative predictive value greater than 99%, similar to LDCT test performance, suggesting the eventuality to use first miRNA tests in this population of patients [29].

In addition to CTC and miRNA, Cohen et al. described the CancerSEEK test, which utilizes combined assays for genetic alterations (mutation present in plasma circulating tumor DNA) and protein biomarkers. Not only does this test have the capacity to identify the presence of stage I to III cancers of the ovary, liver, stomach, pancreas, esophagus, colo-rectum, lung, or breast, but also to localize the organ of origin of these cancers [30].

Although still exploratory, gene expression profiling in the respiratory epithelium may help assess LC risk in the setting of detection of early-stage LC. Indeed, there is some evidence that nontumor adjacent cells share some molecular characteristics with tumor cells. In this context, it has been recently demonstrated that most genetic alterations expressed in smoker patients are not only found in bronchial but also in the nasal epithelium [31]. Thus, LC-associated gene expression assessment in nontumor respiratory epithelium may represent a promising field of development to optimize LC risk evaluation and, thus, to better understand its pathogenesis [31].

#### 2.3.2. Biomarkers to Identify High-Risk Individuals and to Spot the Optimal Target Population for Lung Cancer Screening

Several markers were investigated in large prospective LCS programs, such as the Continuous Observation of Smoking Subjects (COSMOS) and the Multicenter Italian Lung Detection (MILD) trial; in interventional programs, such as the Carotene and Retinol Efficacy Trial (CARET); and in observational cohorts, such as the European Prospective Investigation into Cancer and Nutrition (EPIC) and the Northern Sweden Health and Disease Study (NSHDS) [28,29,32]. The miRNA signature [28,29] and serum proteins [32] were evaluated and performed well in determining a risk score for developing LC.

Other investigations, such as methylation of free plasma DNA, are potential options for identifying individuals at high risk of developing LC [33,34,35,36,37,38] but still need to be evaluated in validation studies.

#### 2.3.3. Biomarkers to Help Classify Indeterminate Lung Nodules

Appropriate management of indeterminate lung nodules is one of the key factors of success in LCS implementation. Several biological signatures have been studied or are under investigation to help classify indeterminate lung nodules. Among them, plasma protein biomarkers combined with clinical risk factors (age, smoking history, nodule diameter, nodule edge characteristics, and nodule location) in an “integrated classifier” performed well in identifying benign nodules among nodules classified as indeterminate [39].

The presence of serum antibodies to a panel of seven LC-associated antigens distinguished malignant from benign nodules in a prospective registry [40]. In this study, patients harboring a 4–20 mm lung nodule with a positive antibodies panel test (EarlyCDT-Lung Test, (ECLS), Oncimmune, De Sotto, MO, USA) had a twofold increased relative risk to develop an LC than patients with a negative test [40]. A combined strategy using ECLS test and risk model integration showed a high specificity (>92%) and a positive predictive value of >70% for LC detection. The capacity of this tumor-associated antigen test, combined with LDCT, to reduce the incidence of late-stage LC at presentation is presently being investigated in a randomized controlled trial [41]. This interventional study includes 12,000 Scottish patients, aged 50–75 years, current or former smokers (with at least 20 pack-years or with less than 20 pack-years plus a family history of LC), tested with ECLS, X-ray chest, and CT scan, and with a follow-up of 24 months [41].

Other immune biological signatures, such as C4d-specific antibodies, have been investigated to diagnose indeterminate lung nodules and showed equivocal results [42].

Finally, a prespecified miRNA signature showed its ability to distinguish LC from the large majority of benign LDCT-detected pulmonary nodules. The combination of MSC/LDCT could reduce LC false-positive rate detection fivefold (19.7% vs. 3.7% for LDCT and MSC/LDCT, respectively) [28]. 

### 2.4. Highly Sensitive Bronchoscopic Techniques to Enhance the Detection Rate of Central Airway Lesions

Low-dose computed tomography has a very low detection rate for central airway lesions that are more commonly squamous cell carcinomas (SqCC) and for preinvasive lesions. Therefore, enhancing LDCT detection rate using bronchoscopic techniques, such as autofluorescence bronchoscopy (AFB), narrow band imaging, or high magnification bronchovideoscopy, can be promising [43]. Some authors have incorporated endoscopic techniques in LCS strategies. McWilliams et al. showed promising results when combining sputum atypia with LDCT and AFB [44,45]. The benefit of combining AFB with LDCT in LCS was not confirmed in the large-scale trial performed by Tremblay et al. [46] in which AFB detected too few CT-occult cancers, and thus failed to show any benefit in high-LC-risk patient screening. Furthermore, due to the decreasing incidence of SqCC, and its precursors, that is, dysplasias and SqCC in situ, relative to adenocarcinoma, and in the absence of a clear survival benefit to detecting precancerous central airway lesions, AFB does not seem to have a place in today’s LCS strategies [47].

## 3. Deep Learning for Early Cancer Detection

As yet, in the setting of early cancer diagnosis, deep learning has essentially been applied to chest imaging interpretation. However, the complexity of the approach of deep learning techniques can now be considered, as soon as one simultaneously analyzes parameters that appear to be completely independent. For instance, when developing their CancerSEEK test, Cohen et al. used supervised machine learning to predict the underlying cancer type. The input algorithm took into account the circulating tumor DNA (ctDNA) and protein biomarker levels as well as the gender of the patient [29].

## 4. Conclusions

It can be reasonably assumed that very soon, European health authorities will provide coverage for LDCT-based LCS. This, added to the global shortage of radiologists, will result in a large number of anxious patients with “indeterminate lung nodules” overloading lung clinics, waiting for repeat chest imaging and invasive tests to obtain a definite answer.

In this context, we strongly believe that there is room for using, as a first reading approach, a CNN-driven LDCT with a predefined detection threshold to label all “nodule-free” examinations as reassuring, and then to incorporate the trilogy of chest imaging, risk factors, and biological signatures into machine learning algorithms to classify the nodules that have been detected according to the level of suspicion. For health care professionals, CNNs are often considered as a black box. Thus, to avoid this pitfall, one will also have to demystify the decision tree and to report on the respective weight of each clinical, radiological, and biological input that led to nodule classification (Figure 6).

## Figures and Tables

**Figure 1 cancers-11-00212-f001:**
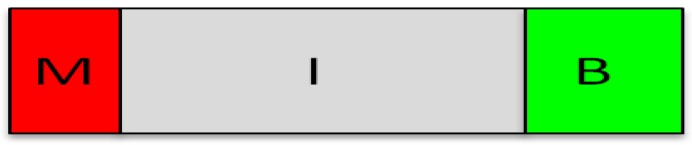
“I” nodules, the “grey zone” of lung cancer screening.

**Figure 2 cancers-11-00212-f002:**
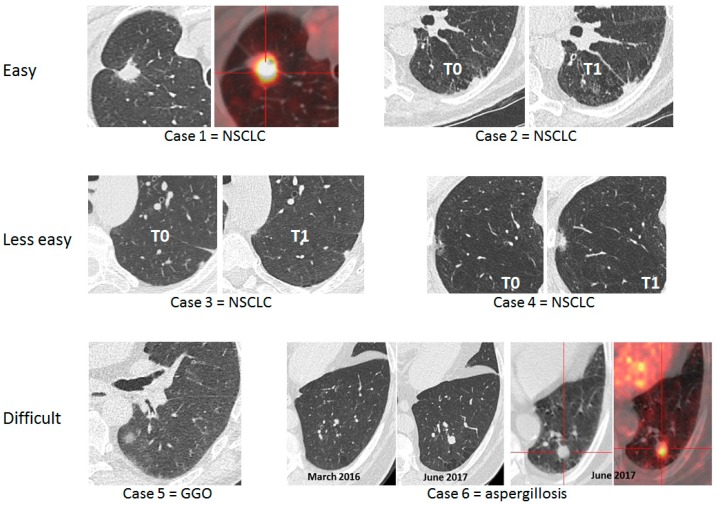
Six lung cancer screening cases illustrating to what extent interpretation of low-dose computed tomography may be difficult. NSCLC: non-small cell lung cancer; GGO: ground glass opacities.

**Figure 3 cancers-11-00212-f003:**
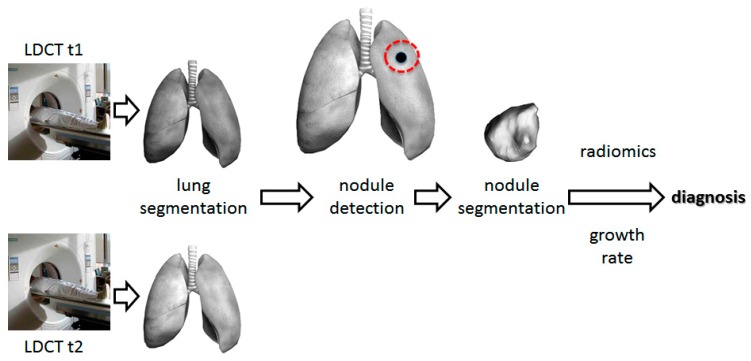
Architecture of computer-aided diagnosis systems for lung cancer screening. t1: first screening round. t2: follow-up.

**Figure 4 cancers-11-00212-f004:**
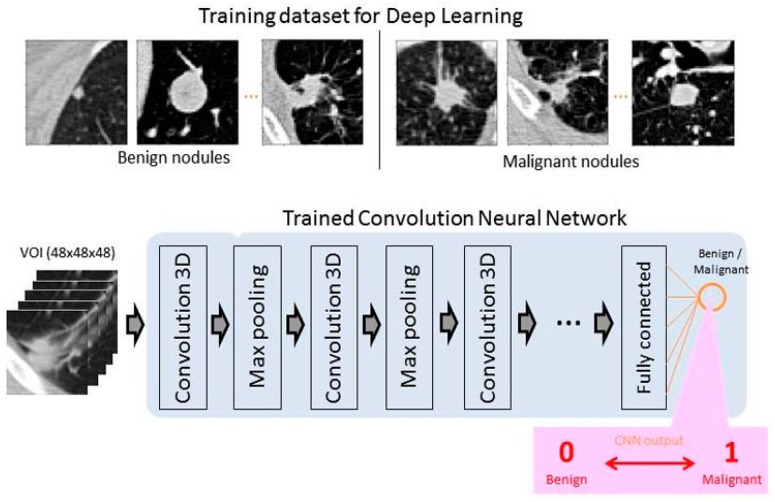
Training CNN for lung cancer screening.

**Figure 5 cancers-11-00212-f005:**
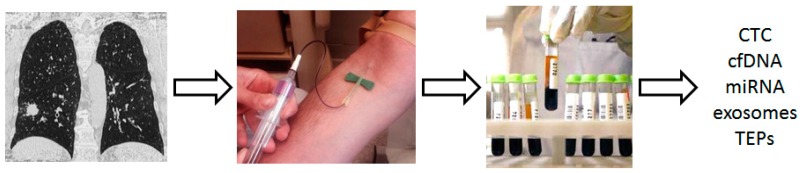
Tumor-derived components that can be used in the setting of lung cancer screening. TEP: tumor-educated platelets; miRNA: microRNA.

**Figure 6 cancers-11-00212-f006:**
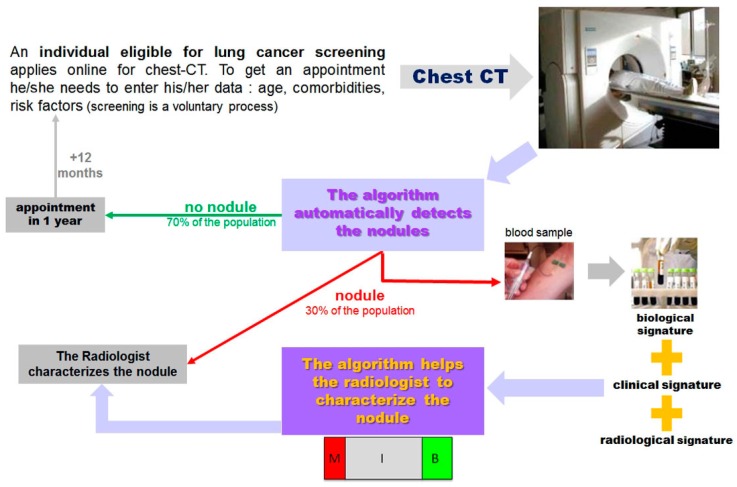
Workflow of lung cancer screening.

**Table 1 cancers-11-00212-t001:** Summary of the National Lung Screening Trial (NLST) and the NELSON trials.

	NLST	NELSON
Country	USA	BE/NL
Enrollment	2002–2004	2003–NR
Number of Centers	33	4
Number of screens		3
Screening planned at years	1, 2 and 3	1, 2 and 4
Comparison	LDCT vs. Xray	LDCT vs. usual care
Population		
Age	55–74	50–69 (50–75)
Smoking (pack-years)	≥30	>15 *
Sex	both (male 59%)	men º (male 84%)
Years since quit	≤15	≤10
Patients Screened, *n*	26,722 vs. 26,732	7907 vs. 7915
Planned follow-up, y	>7	10
Nodule Size warranting Follow-up	2011	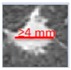	2009	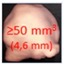	+ VDT
2014	≥100 mm^3^(≥5 mm)	+ VDT
LC diagnosed at screening, %	1.02	0.9
5 mm Reduction of LC mortality	20%	26% ^a^

*, ≥15 cigarettes/day for 25 years or ≥10 cigarettes/day for 30 years; º, both in Belgium; VDT, volume doubling time; ^a^, in men.

**Table 2 cancers-11-00212-t002:** Publicly available databases for lung cancer screening.

	Kaggle	Luna16	NLST	COPDG Gene	LTRC
**Number**	1397	888	>1000	>1000	>1000
**Date**			2002–2004 to 2009	Start 2008	
**Available without registration**	Yes	Yes	No	No	No
**COPD * cases**				Yes	Yes
**Ground truth**	Cancer/no cancer one year after the CT scan	x, y, z coordinates and diameter of nodules			
**Image data**	Yes	Yes	Yes	Yes	Yes
**Cohort level**	“high-risk patient”		Age 55–74 >30 years smoking history <15 years since quitting	Age 45–80 >10 pack-years smoking history	“Most donor subjects have interstitial fibrotic lung disease or COPD” Average age 60
**Individual level**			Questionnaire: living condition, family history. Cancer diagnosis: location/tumor size	Subject phenotype: living condition, gender, medical history, comorbidities, physical characteristics…	Clinical and pathological diagnoses, pulmonary function tests, living condition, exercises tests…
**Biological data**			Lung tissues	SNP genotype	Blood and lung tissues

* Chronic obstructive pulmonary disease (COPD); CT: computed tomography; SNP: single nucleotide polymorphisms.

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
