# Peer review of "Lung Cancer Screening, towards a Multidimensional Approach: Why and How?"

_cancers, 2019, doi:10.3390/cancers11020212_

Round 1
Reviewer 1 Report
1. Authors should recheck grammar and type mistakes. There are several mistakes.
For example, in line 69, “ [NNS]” should be “(NNS)”.
2. In table 1, please make sure the percentage of LC diagnosed at screening of NELSON is “0, 9”, but not “0.9”.
3. In table 1, please make sure “4 (1,2,2)” is correct (Nb of screens planed). Please clearly explain the means of “(1, 2, 2)”.
4. This sentence can cause some confusion. Therefore, I think authors should revise it. Readers may think that you mean “1972 teams”? Please also add related references.
“Using a data set of thousands of high-resolution lung scans provided by the 121 National Cancer Institute, 1972 teams developed algorithms to accurately determine when lesions in 122 the lungs are cancerous.”
5. Please add more detail information in those content related to the following references: 18 (line 121, 128), 27, 28 (line 151), 30 (line 161), 39, 40.
6. Authors should point out the full name at the first appearance of initials word. For example, the full name of COPD (Chronic obstructive pulmonary disease) is not included in this manuscript.
7. In figure 3, the graph of case 5 and graphs of case 6 are too close to each other, which may cause misunderstanding.
Author Response
Reviewer 1 Comments and Suggestions for Authors 1. Authors should recheck grammar and type mistakes. There are several mistakes. For example, in line 69, “ [NNS]” should be “(NNS)”. Author: This has now been fixed 2. In table 1, please make sure the percentage of LC diagnosed at screening of NELSON is “0, 9”, but not “0.9”. Author: This has now been fixed 3. In table 1, please make sure “4 (1,2,2)” is correct (Nb of screens planed). Please clearly explain the means of “(1, 2, 2)”. Author: This has now been checked and fixed 4. This sentence can cause some confusion. Therefore, I think authors should revise it. Readers may think that you mean “1972 teams”? Please also add related references. “Using a data set of thousands of high-resolution lung scans provided by the 121 National Cancer Institute, 1972 teams developed algorithms to accurately determine when lesions in 122 the lungs are cancerous.” Author: This sentence now appears as “Using a data set of 2101 high-resolution lung scans provided by the National Cancer Institute and labelled as “with or without cancer”, the 1972 competing teams have developed algorithms to accurately determine when lesions in the lungs were cancerous” and the related reference has been added 5. Please add more detail information in those content related to the following references: 18 (line 121, 128), 27, 28 (line 151), 30 (line 161), 39, 40. Author: This has now been done 6. Authors should point out the full name at the first appearance of initials word. For example, the full name of COPD (Chronic obstructive pulmonary disease) is not included in this manuscript. Author: This has now been fixed 7. In figure 3, the graph of case 5 and graphs of case 6 are too close to each other, which may cause misunderstanding. Author: The figure has been modified accordingly
Reviewer 2 Report
This review deals with lung cancer screening and summarized 3 ways to improve the screening based on a literature review. These include (1) refining risk factor assessment, (2) using computer aided diagnosis (deep learning neural network), and (3) blood biomarker analysis. The paper is well structured and well written and should be interesting to the readers of Cancers.
The paper is focused on LDCT, which is more useful for periphery lung cancer detection and this is well dealt in the manuscript. The role of LDCT for central airway lung cancer detection, its pros and corns should also be added to give a complete picture of the lung cancer screening issue. The follow-up for periphery and central airway lung cancers are also different and again the manuscript did a great job on the periphery cancers, but did not talk about much on the central airway lesion follow-up. Adding related content and the role of bronchoscopy technologies for the central airway lesion management is necessary for presenting a complete picture to the readers.
Minor issues:
There are a few places where the bulletin number 1), 2), 3) etc. are shown as 1/, 2/, 3/. This should be fixed.
Copyright permissions should be obtained for figures taken from other publications.
Author Response
Reviewer 2 Comments and Suggestions for Authors This review deals with lung cancer screening and summarized 3 ways to improve the screening based on a literature review. These include (1) refining risk factor assessment, (2) using computer aided diagnosis (deep learning neural network), and (3) blood biomarker analysis. The paper is well structured and well written and should be interesting to the readers of Cancers. The paper is focused on LDCT, which is more useful for periphery lung cancer detection and this is well dealt in the manuscript. The role of LDCT for central airway lung cancer detection, its pros and corns should also be added to give a complete picture of the lung cancer screening issue. The follow-up for periphery and central airway lung cancers are also different and again the manuscript did a great job on the periphery cancers, but did not talk about much on the central airway lesion follow-up. Adding related content and the role of bronchoscopy technologies for the central airway lesion management is necessary for presenting a complete picture to the readers. Author: A chapter titled “2.4. Highly sensitive bronchoscopic techniques to enhance the detection rate of central airway lesions” has now been added to the this manuscript in order to deal with the central airway lesions detection issue Minor issues: There are a few places where the bulletin number 1), 2), 3) etc. are shown as 1/, 2/, 3/. This should be fixed. Author: This has now been fixed Copyright permissions should be obtained for figures taken from other publications. Author: Figure 4 which came from reference 14 has now been deleted
Round 2
Reviewer 1 Report
The authors already adequately address my questions and concerns. I don't have other questions. Please double double check the language (grammar, spell check, et. al).